# Management Pathways for Traumatic Rib Fractures—Importance of Surgical Stabilisation

**DOI:** 10.3390/healthcare11081064

**Published:** 2023-04-07

**Authors:** Selwyn Selvendran, Rajkumar Cheluvappa

**Affiliations:** 1Department of Surgery, St George Hospital, Kogarah, NSW 2217, Australia; tselvendran@hotmail.com; 2Nursing and Midwifery, Australian Catholic University, Watson, ACT 2602, Australia

**Keywords:** chest trauma, flail chest, randomised control trial, rib fracture, surgical stabilisation of rib fracture, thoracic cage, thoracic injury

## Abstract

Rib fractures occur in almost half of blunt chest wall trauma victims in Australia. They are associated with a high rate of pulmonary complications, and consequently, with increased discomfort, disability, morbidity, and mortality. This article summarises thoracic cage anatomy and physiology, and chest wall trauma pathophysiology. Institutional clinical strategies and clinical pathway “bundles of care” are usually available to reduce mortality and morbidity in patients with chest wall injury. This article analyses multimodal clinical pathways and intervention strategies that include surgical stabilisation of rib fractures (SSRF) in thoracic cage trauma patients with severe rib fractures, including flail chest and simple multiple rib fractures. The management of thoracic cage injury should include a multidisciplinary team approach with proper consideration of all potential avenues and treatment modalities (including SSRF) to obtain the best patient outcomes. There is good evidence for the positive prognostic role of SSRF as part of a “bundle of care” in the setting of severe rib fractures such as ventilator-dependent patients and patients with flail chest. However, the use of SSRF in flail chest treatment is uncommon worldwide, although early SSRF is standard practice at our hospital for patients presenting with multiple rib fractures, flail chest, and/or severe sternal fractures. Several studies report that SSRF in patients with multiple simple rib fractures lead to positive patient outcomes, but these studies are mostly retrospective studies or small case–control trials. Therefore, prospective studies and well-designed RCTs are needed to confirm the benefits of SSRF in patients with multiple simple rib fractures, as well as in elderly chest trauma patients where there is scant evidence for the clinical outcomes of SSRF intervention. When initial interventions for severe chest trauma are unsatisfactory, SSRF must be considered taking into account the patient’s individual circumstances, clinical background, and prognostic projections.

## 1. Introduction

### 1.1. History of Rib Fractures and Surgical Stabilisation of Rib Fractures (SSRF)

The Greek physician Soranus first described the pathophysiology of rib fracture in the early first century, but only in the 1940s and 1950s was surgical treatment first advocated [1]. However, this fell out of favour when intermittent positive pressure ventilation (IPPV) became available to treat patients with respiratory failure due to severe rib fracture of the flail segment and complications. The specifics of SSRF, its prognostic benefits, and its impact on clinical progression (using a cohort of patients on mechanical ventilation) were first published in the 1970s [2]. However, pulmonary physiology and the biomechanical function of the chest wall were only much better understood during the 1980s [3]. Moreover, the quality of appliances used in SSRF improved dramatically during the 1980s. Over the last decade or so, multiple studies have indicated that SSRF as part of an official, organised, multimodal pathway could improve the outcome for patients with severe rib fractures and flail chests [4,5,6]. However, major centres are still reluctant to undertake SSRF for patients with flail chest and multiple rib fractures. Moreover, there is inadequate level I or II evidence for SSRF in non-ventilator-dependent patients with simple rib fractures.

### 1.2. Anatomy of the Thoracic Cage and the Physiology of Breathing

It is important to understand the anatomy and physiology of the thoracic cage (rib cage) and of its contents towards comprehending the pathophysiology of rib fractures. The thoracic cage consists of a musculoskeletal framework that is held together by 12 thoracic vertebrae [7]. They form the ribs that surround the thoracic cavities and attach to the manubrium and sternum anteriorly [7]. The first seven ribs attach to the sternum by costal cartilages and are known as true ribs [8]. The remaining ribs are known as false ribs, where the 8th, 9th, and 10th ribs are connected to the 7th costal cartilage; and the 11th and 12th ribs are free-floating ribs that end in the muscle wall [8]. The mediastinum and two pleural cavities on each side separate the thoracic cavity into three anatomical compartments [9]. The heart and major vessels are contained in the mediastinum, whereas the lungs are contained in the pleural cavities on either side [7]. The thoracic cage protects the lungs and heart, and provides attachments for the muscles of the thorax, upper extremity, back, and abdomen. The thoracic cage connects to the neck via the thoracic inlet, and the muscular diaphragm separates it from the abdomen inferiorly [9].

The twelve ribs in the musculoskeletal chest wall function as a unit alongside the diaphragm to facilitate ventilation [8]. During inspiration, the diaphragm contracts and the intercostal muscles increase the intrathoracic volume. This lowers intrathoracic pressure and permits passive airflow into the lungs [10]. During expiration, the diaphragm relaxes and the intercostal muscles actively force air out of the lungs [10]. This coordinated lung physiology is disrupted by injuries to the chest wall.

### 1.3. Pathophysiology of Thoracic Trauma Sequela

Thoracic trauma causes disruption to the chest wall, impacting respiration and/or circulation, leading to death or increased morbidity. Rib fractures can compromise respiratory function by direct pulmonary injury and/or by interfering with breathing mechanics. Direct trauma to the lungs or central circulatory system can cause immediate life-threatening conditions such as tension pneumothorax, cardiac tamponade, aortic injury, haemothorax, and tracheobronchial disruption [11]. Moreover, thoracic injuries such as pulmonary contusions and other lung injuries often lead to ventilation–perfusion mismatch and reduced lung compliance, leading to hypoventilation/hypoxia requiring intubation, IPPV, and intensive care unit (ICU) admission [10,12].

Rib fractures complicating trauma are common in Australia, occurring in 20% to 40% of blunt chest wall trauma patients [13]. Rib fractures are also associated with other concomitant injuries, multiple complications, and poor outcomes [14]. Pain, atelectasis, lobar collapse, pulmonary effusion, aspiration, pulmonary embolism, adult respiratory distress syndrome, and pneumonia are some of the thoracic complications associated with rib fractures [15].

In elderly patients with osteoporosis or brittle bones, rib fractures are associated with relatively higher mortality and morbidity [16]. Patients with flail chest have increased risk of other complications. Underlying lung injuries result in further contusion with pain and splinting of the chest wall resulting in pneumonia and respiratory failure [12]. Therefore, it is important to have a management plan/pathway for bad rib fractures and high-risk populations; a plan/pathway that includes strategic pain control and surgical stabilisation of rib fractures (SSRF).

Despite evidence and benefits, a surgical approach is still not widely utilised in major metropolitan trauma centres in Australia. The aim of this paper is to review and evaluate the pathophysiology of rib fractures and the outcome-based role of SSRF as part of an institutional clinical plan/pathway.

## 2. Materials and Methods

Extensive searches were conducted using PubMed, Google Scholar, and government reports. Word searches were performed with relevant combinations of words such as chest trauma, chest injury, flail chest, randomised control trial, observation study, retrospective study, cross-sectional study, rib fracture; surgical stabilisation of rib fracture, thoracic injury, video-assisted thoracoscopic surgery, and outcomes. From more than 70 references, we identified ~32 papers that addressed management pathways for acute traumatic rib fractures involving surgical stabilisation. References from publications resulting from these sources (~6 papers) were also investigated for relevant data or information suitable for our manuscript. Topically relevant general review articles were excluded. Original clinical research, systematic reviews, and meta-analyses demonstrating statistical significance were dissected further with reference to criteria and outcomes involving continuous positive airway pressure (CPAP), bilevel positive airway pressure (BiPAP), chest injury activation protocol (ChIP), high flow nasal prongs (HFNP), hospital length of stay (HLOS), ICU length of stay (ILOS), intermittent positive pressure ventilation (IPPV), mechanical ventilator days (MVDs). This honing approach resulted in ~36 publications that informed the evidence behind our report.

Ethical approval was not required by our institution for this submission. However, in several (latter) parts of this paper, we include observations and preliminary concepts/results from our use of SSRF in our trauma care practice at our hospital. Moreover, we include information about ongoing clinical studies linked to the information presented in this paper. Unfortunately, this paper does not have an adequate word count to fit into a submission under the “review” category. Therefore, this paper has been processed as a “brief report”.

This submission did not use the preferred reporting items for systematic reviews and meta-analysis (PRISMA) as it was not relevant. However, it is evident that we utilised a systematic approach for this submission. Moreover, we are currently preparing a systematic PRISMA-guided systematic review covering evidence-based management for elderly patients with rib fractures. This manuscript, which we expect to be submitted in a few weeks’ time, may have an adequate word count to fit into the “review category” as a systematic review.

## 3. A complete “Bundle of Care” for Rib Fractures

Complications such as pneumonia and respiratory failure are common when blunt chest injuries are not treated promptly with adequate analgesics, physiotherapy, and respiratory support [17]. These problems are often associated with a delayed recovery, prolonged hospital length of stay (HLOS), mechanical ventilator days (MVD), ICU length of stay (ILOS), increased mortality, chronic pain, pulmonary complications, reduced quality of life, and increased absorption of limited public resources [17]. In adults over the age of 65, rib fractures are the most common clinical fracture associated with mortality and morbidity. Increased fractures (≥3), frailty, and age (≥65 years) are associated with increased hospital mortality and morbidity [16].

Delayed and inadequate analgesia for chest trauma patients can lead to anorexia, insomnia, stress, and limited mobility [18]. These promote inadequate breathing, coughing, and activity engagement [18]. Patients often present with isolated blunt chest wall injury from low-energy mechanisms such as falling from standing heights [19]. Unfortunately, these patients do not satisfy trauma team activation criteria for receiving multidisciplinary review. Curtis et al. set up a multidisciplinary review pathway called the chest injury activation protocol (ChIP), which responds promptly to blunt chest injuries with incentive spirometry, physiotherapy, oxygen via high flow nasal prongs (HFNP), and multimodal analgesia including patient-controlled analgesia (PCA) [20]. In this pre–post cohort study, the ChIP pathway was shown to reduce the incidence of pneumonia in patients with blunt chest injuries [20].

Several studies show standardised clinical pathways that streamline patients with rib fractures and improves clinical outcomes for chest trauma patients. Sahr et al. set up a “bundle of care”, which included early referral to trauma services [21]. This pathway showed improved outcomes including shorter HLOS and ILOS [21]. Todd et al. showed that the introduction of their clinical “bundle of care” decreased MVD, HLOS, infectious morbidity, and mortality in high-risk trauma patients [22]. The Eastern Association for the Surgery of Trauma and Trauma Anaesthesiology Society guidelines recommend a multimodal approach to analgesia, including PCA and regional blocks [6]. When adequate pain control is in place, early mobilisation, pulmonary hygiene, and early physiotherapy intervention as part of an organised “bundle of care” improves the outcome for these patients [6]. Therefore, a comprehensive, streamlined, and standardised institutional-based multimodal pathway (“bundle of care”) improves outcomes for patients with rib fractures [5].

## 4. The Role of Surgical Stabilisation of Rib Fractures (SSRF) in Management Pathways for Chest Trauma with Rib Fractures

As part of these pathways or bundles of care, SSRF should also be considered. Early SSRF (≤72 h) is now advocated in our hospital for patients meeting set criteria such as multiple rib fractures, flail chest, trauma patients requiring video-assisted thoracoscopic surgery (VATS), severe sternal fractures, and extubation failure. Unfortunately, the use of SSRF in flail chest treatment is still uncommon worldwide. One study showed only 0.7% of 3467 adults with post-trauma flail chest undergoing SSRF at level 1 and 2 trauma centres [23].

In the absence of flail chest in patients with multiple rib fractures, several papers report the successful use of SSRF with positive clinical outcomes [24,25,26,27]. However, prospective studies are conspicuously absent that can objectively prove this therapeutic modality as authoritative in patients with multiple rib fractures.

## 5. Flail Chest and Its Management

Developing a flail chest from blunt trauma can be a life-threatening injury. A flail chest is characterised by paradoxical motion of the flailing chest wall segment that results from three or more successive rib fractures in two or more places. The considerable morbidity and high mortality rate associated with flail chest varies between 20% and 30% [2]. Flail chest also leads to chronic discomfort, disability, and loss of working days. The treatment for flail chest differs depending on the severity of the injury, number of ribs involved and other underlying comorbidities [28]. The usual management for flail chest is admission to ICU, with strong analgesia, physiotherapy, respiratory toileting, and non-invasive respiratory support such as HFNP, IPPV, continuous positive airway pressure (CPAP), and bilevel positive airway pressure (BiPAP) in deteriorating patients [28,29].

## 6. The role of Surgical Stabilisation of Rib Fractures (SSRF) in Flail Chest Management

Current evidence suggests that conservative treatment alone may not guarantee a positive outcome [12]. SSRF is becoming more popular. There have been multiple randomised control trials (RCTs) and meta-analyses carried out on SSRF on flail chests [28].

A meta-analysis of pooled data from 11 studies showed SSRF to be beneficial in flail chest patients resulting in shorter mechanical ventilator days (MVDs), hospital length of stay (HLOS), ICU length of stay (ILOS), chest deformity, pneumonia incidence, septicaemia, need for tracheostomy, and mortality [30]. A meta-analysis of nine studies showed SSRF decreased MVDs, HLOS, ILOS, pneumonia incidence, need for tracheostomy, and mortality in flail chest patients [15]. A 2019 RCT conducted by Liu et al. with 53 patients showed that SSRF lowered pneumonia incidence, adult respiratory distress syndrome, thoracic deformity, and pain [31]. In the same study, patients who underwent SSRF had shorter MVDs and ILOS but no difference in HLOS [31]. A 2002 case–control study by Tanaka et al. on 37 patients also showed that SSRF lowered MVD, ILOS, and other complication rates [32]. This study also revealed that conducting SSRF in flail chest patients improved forced vital capacity (FVC), lung function, and cost-effectiveness; and ensured the quicker return of patients to fulltime work [32]. A 2020 observational study by Wu et al. showed that ventilator-dependent patients with acute respiratory failure after severe blunt chest injury who underwent SSRF had shorter MVDs, HLOS, and pneumonia incidence [33].

A 2020 meta-analysis by Long et al. scanned 1917 articles and identified 7 eligible RCTs, with a total of 538 multiple rib fracture patients for their analysis [4]. Out of this large sample size of 538 patients, 260 underwent SSRF, and 278 underwent non-surgical treatment [4]. A reduction in complications such as pneumonia, pain, and chest wall deformity were evident in patients who underwent SSRF [4]. The utilization of SSRF also reduced MVDs, ILOS, and unlike Liu et al., HLOS as well [4,31]. This study also showed decreased medical expenses in patients who underwent SSRF [4].

Currently, SSRF for flail chest patients is recommended by the British National Institute for Health and Clinical Excellence (NICE) and the American Eastern Association for the Surgery of Trauma (EAST) to decrease MVDs, HLOS, ILOS, pneumonia incidence, and mortality [34].

## 7. The Role of Surgical Stabilisation of Rib Fractures (SSRF) in Simple Rib Fracture Management

Management approaches encompassing SSRF are also being evaluated for simple rib fracture patients. The results of a multicentric RCT carried out in level 1 trauma centres in metropolitan Australia were recently published by Marasco et al. [35]. Although the study involved non-ventilator-dependent patients with multiple rib fractures, the design of the study with a large double-crossover design makes the inconclusive nature of the SSRF-related outcomes irrelevant.

In a 2019 cohort study paper involving non-ventilator-dependent patients with multiple simple rib fractures, Lin, et al. showed that adding SSRF during VATS improves patient outcomes such as better pain control, and decreased ILOS and HLOS [36]. In a 2012 paper, Girsowicz et al. reviewed 356 papers and selected 9 retrospective and case–control studies that showed SSRF was safe in simple rib fractures, and improved patient outcomes involving recovery time, pain management, respiratory function, and quality of life [26]. Studies by Nirula et al. (2006), Richardson et al. (2007), Campbell et al. (2009), and Pieracci et al. (2022) have also shown that SSRF improves patient outcomes in the setting of multiple simple rib fractures [24,25,27,37].

## 8. Strengths of, and New Information from Our Brief Report

We analysed current thoracic trauma management algorithms (“bundles of care”) and evidence-based rib fracture therapy interventions (including SSRF), which focus on pain control, injury seriousness, and the anatomical extent of involvement. There is good evidence for the prognostic role of SSRF as part of a “bundle of care” in the setting of severe rib fractures such as ventilator-dependent patients and patients with flail chest. At our hospital, early SSRF (within 3 days) is recommended for patients with multiple rib fractures, flail chest, severe sternal fractures, extubation failure, and trauma patients requiring VATS.

We showed that blunt chest wall trauma and rib fractures are common in the elderly, with its associated mortality and morbidity far higher in the elderly when compared to that in younger patients [38]. Currently, the Gold Coast University Hospital is planning a multicentre RCT involving elderly patients with blunt chest wall trauma and rib fractures; and is recruiting investigators Australia-wide. Under current guidelines, pneumonia and contaminated fields are contraindications to SSRF. However, we have successfully conducted SSRF in several multi-trauma patients with contaminated wound fields and pneumonia.

## 9. Weaknesses and Research Gaps Identified in Our Brief Report

Although good quality evidence exists for the utility of SSRF in patients with severe rib fractures and/or flail chest, we also reveal that SSRF use in flail chest treatment is minimal (0.7%) worldwide in these clinical settings [23]. Although several papers report that the use of SSRF in patients with multiple simple rib fractures leads to positive patient outcomes, the evidence from these papers comes mostly from retrospective and case–control trials, and better quality evidence is required to substantiate this approach [24,25,26,27].

Although blunt chest wall trauma and rib fractures are common in the elderly, and their associated mortality and morbidity are far higher, evidence for the clinical outcomes of SSRF in elderly rib fracture patients is sparse (anecdotal) or non-existent. This underlines the necessity for conducting RCTs and prospective studies towards establishing evidence-based guidelines to manage this frail group of patients.

Although we presented the first ever retrospective study on outcomes of SSRF in multi-trauma patients with contaminated wound fields and positive sputum culture, our monocentric study had a small sample size.

## 10. “Take-Home” Points from Our Brief Report

The clinical impact of thoracic trauma depends on the severity of the trauma, the anatomical regions involved, and the age of patients.When initial interventions are unsatisfactory, VATs and SSRF must be considered considering the patient’s unique individual/holistic circumstances, clinical background, and prognostic expectations.Trauma centres and other healthcare providers should have updated evidence-based information about available management modalities. The best outcomes for individual patients are accomplished by combining various treatment approaches and individualising them according to the patient’s holistic situation and clinical condition.As part of rib fracture management pathways or “bundles of care”, early SSRF must be considered.Prospective studies and well-designed RCTs are required to confirm the benefits of SSRF in multiple simple rib fractures and in elderly patients with any sort of rib fracture(s).At our hospital, early SSRF is standard procedure for patients with multiple rib fractures, flail chest, severe sternal fractures, extubation failure, and trauma patients requiring VATS.In 2019, we presented the encouraging results of a small retrospective study on SSRF outcomes in multi-trauma patients with contaminated wound fields and positive sputum culture.

## 11. Conclusions

The standard management of thoracic injury includes a multidisciplinary team approach with proper consideration of all potential avenues and treatment modalities (including SSRF) to give the best possible clinical outcome to the patient. Trauma centres and other healthcare providers should have updated evidence-based information about available management modalities. Multiple randomised control trials (RCTs) and meta-analyses show SSRF to be beneficial in flail chest patients resulting in reduced mortality and morbidity including shorter MVDs, HLOS, and ILOS; and a lower incidence of infections, pneumonia, sepsis, and other complications. Prospective studies and well-designed RCTs are badly needed to confirm the benefits of SSRF in multiple simple rib fractures and in elderly patients with any sort of rib fracture. Such quality studies with good sample sizes are also essential to evaluate the outcome of SSRF in patients diagnosed with pneumonia or contaminated surgical fields.

## Data Availability

Not applicable.

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
