# Peer review of "Management Pathways for Traumatic Rib Fractures—Importance of Surgical Stabilisation"

_healthcare, 2023, doi:10.3390/healthcare11081064_

Round 1
Reviewer 1 Report (Previous Reviewer 1)
thanks for your revision
Author Response
REVIEWER 1
thanks for your revision
You are welcome – Thank you for reviewing our submission positively.
Reviewer 2 Report (Previous Reviewer 2)
The authors modified the paper according to the recommendations. The paper is recommended for the publication in the present version.
Author Response
REVIEWER 2
The authors modified the paper according to the recommendations. The paper is recommended for the publication in the present version.
- Thank you very much for your encouraging comments and positive assessment of our submitted manuscript.
Reviewer 3 Report (Previous Reviewer 3)
According to the author's guidelines of Healthcare, "Brief reports are short, observational studies that report preliminary results or a short complete study or protocol. Brief reports usually contain two figures and/or a table; however, the Materials and Methods sections should be detailed to ensure reproducibility of the presented work. The structure is similar to that of an article, and there is a suggested minimum word count of 2500 words.".
This article is a short, non-systematic, narrative review, not a brief report.
Moreover, the authors did not provide any new insights.
These issues cannot be fixed with a simple revision, I hence recommend rejection.
Author Response
REVIEWER 3
According to the author's guidelines of Healthcare, "Brief reports are short, observational studies that report preliminary results or a short complete study or protocol. Brief reports usually contain two figures and/or a table; however, the Materials and Methods sections should be detailed to ensure reproducibility of the presented work. The structure is similar to that of an article, and there is a suggested minimum word count of 2500 words.". This article is a short, non-systematic, narrative review, not a brief report. Moreover, the authors did not provide any new insights. These issues cannot be fixed with a simple revision, I hence recommend rejection.
- This Reviewer’s feedback is a rehash of his /her previous review which was, “Dear authors, "brief report" is described as follows - "Brief reports are short, observational studies that report preliminary results or a short complete study or protocol. Brief reports usually contain two figures and/or a table; however, the Materials and Methods sections should be detailed to ensure reproducibility of the presented work." (check https://www.mdpi.com/about/article_types for further information) The paper does not present any preliminary results and does not have any materials and methods section. In my opinion, it cannot be considered a "brief report". This kind of issue can not be solved with a simple revision, that's why I must recommend rejection.”
- There is essentially no difference in matter/substance between this reviewer’s first review of our manuscript and his/her second review of our manuscript.
- All “issues” raised by this reviewer have been thoroughly addressed in our previous response as follows:
- We have re-written the manuscript according to the instructions for “Brief Reports” in
- https://www.mdpi.com/journal/healthcare/instructions and
- https://www.mdpi.com/about/article_types
- We have added 5 new sections to our manuscript. One of these sections addresses this reviewer’s (Reviewer 3) concerns as follows -
- We have re-written the manuscript according to the instructions for “Brief Reports” in
“Materials and methods
Extensive searches were conducted using PubMed, Google Scholar, and government reports. Word searches with relevant combinations of words like chest trauma, chest injury, flail chest, randomised control trial, observation study, retrospective study, cross-sectional study, rib fracture; surgical stabilisation of rib fracture, thoracic injury, video-assisted thoracoscopic surgery, and outcomes. From more than 70 references, we identified ~32 papers which addressed management pathways for acute traumatic rib fractures involving surgical stabilisation. References from publications resulting from these sources (~6 papers) were also investigated for relevant data or information suitable for our manuscript. Topically relevant general review articles were excluded. Original clinical research, systematic reviews, and meta-analyses demonstrating statistical significance were dissected further with reference to criteria and outcomes involving continuous positive airway pressure (CPAP), bilevel positive airway pressure (BiPAP), chest injury activation protocol (ChIP), high flow nasal prongs (HFNP), hospital length of stay (HLOS), ICU length of stay (ILOS), intermittent positive pressure ventilation (IPPV), mechanical ventilator days (MVDs). This honing approach resulted in ~36 publications which informed the evidence behind our report.
Ethical approval was not required by our institution for this submission. However, in several (latter) parts of this paper, we have included observations and preliminary concepts/results from our use of SSRF in our trauma care practice at our hospital. Moreover, we have also included information about ongoing clinical studies linked to the information presented in this paper. Unfortunately, this paper does not have an adequate word count to fit into a submission under the “review” category. Therefore, this paper has been processed as a “brief report”.
This submission did not use the preferred reporting items for systematic reviews and meta-analysis (PRISMA) as it was not relevant. However, it is evident that we utilised a systematic approach for this submission. Moreover, we are currently preparing a systematic PRISMA-guided systematic review covering evidence-based management for elderly patients with rib fractures. This manuscript, which we expect to be submitted in a few weeks’ time, may have an adequate word count to fit into the “review category” as a systematic review.”
- Therefore, all “issues” raised by this reviewer have already been thoroughly, correctly, and adequately addressed in our previous response.

Round 2
Reviewer 3 Report (Previous Reviewer 3)
Dear authors,
the study would not be an observational study, even if you wrote and rewrote it. There is no difference between the first and the second round of review because the study has got design issues which you can not address by adding a paragraph. Moreover, in the paragraph headed "materials and methods", you basically reported the screening process of your review. While the materials and methods paragraph of an observational study (such as the brief report) is supposed to show: the population and context, sampling method, measurement and outcomes, and statistical analysis. Hence, the paragraph headed "materials and methods" is not a material and methods paragraph. Finally, as you state in materials and methods, is a short narrative review, not a brief report and I will recommend rejection.
Author Response
This Reviewer’s feedback is a rehash of his /her previous review which was, “Dear authors, "brief report" is described as follows - "Brief reports are short, observational studies that report preliminary results or a short complete study or protocol. Brief reports usually contain two figures and/or a table; however, the Materials and Methods sections should be detailed to ensure reproducibility of the presented work." (check https://www.mdpi.com/about/article_types for
further information) The paper does not present any preliminary results and does not have any
materials and methods section. In my opinion, it cannot be considered a "brief report". This kind of issue can not be solved with a simple revision, that's why I must recommend rejection.” There is essentially no difference in matter/substance between this reviewer’s first review of our manuscript and his/her second review of our manuscript. All “issues” raised by this reviewer have been thoroughly addressed in our previous response as follows:
o We have re-written the manuscript according to the instructions for “Brief Reports” in
https://www.mdpi.com/journal/healthcare/instructions and
https://www.mdpi.com/about/article_types
o We have added 5 new sections to our manuscript. One of these sections addresses this reviewer’s (Reviewer 3) concerns as follows -
“Materials and methods
Extensive searches were conducted using PubMed, Google Scholar, and government reports. Word searches with relevant combinations of words like chest trauma, chest injury, flail chest, randomised control trial, observation study, retrospective study, cross-sectional study, rib fracture; surgical stabilisation of rib fracture, thoracic injury, video-assisted thoracoscopic surgery, and outcomes. From more than 70 references, we identified 32 papers which addressed management pathways for acute traumatic rib fractures involving surgical stabilisation. References from publications resulting from these sources (6
papers) were also investigated for relevant data or information suitable for our manuscript. Topically relevant general review articles were excluded. Original clinical research, systematic reviews, and meta-analyses demonstrating statistical significance were dissected further with reference to criteria and outcomes involving continuous positive airway pressure (CPAP), bilevel positive airway pressure (BiPAP), chest injury activation protocol (ChIP), high flow nasal prongs (HFNP), hospital length of stay (HLOS), ICU length of stay (ILOS), intermittent positive pressure ventilation (IPPV), mechanical ventilator days (MVDs). This honing approach resulted in 36 publications which informed the evidence behind our report. Ethical approval was not required by our institution for this submission. However, in several (latter) parts of this paper, we have included observations and preliminary concepts/results from our use of SSRF in our trauma care practice at our hospital. Moreover, we have also included information about ongoing clinical studies linked to the information presented in this paper. Unfortunately, this paper does not have an adequate word count to fit into a submission under the “review” category. Therefore, this paper has
been processed as a “brief report”. This submission did not use the preferred reporting items for systematic reviews and metaanalysis (PRISMA) as it was not relevant. However, it is evident that we utilised a systematic approach for this submission. Moreover, we are currently preparing a systematic PRISMA-guided systematic review covering evidence-based management for elderly patients with rib fractures. This manuscript, which we expect to be submitted in a few
weeks’ time, may have an adequate word count to fit into the “review category” as a systematic review.” Therefore, all “issues” raised by this reviewer have already been thoroughly, correctly, and adequately addressed in our previous response.

This manuscript is a resubmission of an earlier submission. The following is a list of the peer review reports and author responses from that submission.
Round 1
Reviewer 1 Report
please form the brief report according to the standard format of the journal
what is the take home massage from your brief report?
What are the strengths and weaknesses of your brief report?
Author Response
REVIEWER 1
please form the brief report according to the standard format of the journal
- Our “Responses to reviewers’ comments” have different citation numbers and order when compared to that of our revised manuscript submission.
- We have re-written the manuscript according to the instructions for “Brief Reports” in https://www.mdpi.com/journal/healthcare/instructions
- We have added 5 new sections to our manuscript. One of these sections is as follows -
“Materials and methods
Extensive searches were conducted using PubMed, Google Scholar, and government reports. Word searches with relevant combinations of words like chest trauma, chest injury, flail chest, randomised control trial, observation study, retrospective study, cross-sectional study, rib fracture; surgical stabilisation of rib fracture, thoracic injury, video-assisted thoracoscopic surgery, and outcomes. From more than 70 references, we identified ~32 papers which addressed management pathways for acute traumatic rib fractures involving surgical stabilisation. References from publications resulting from these sources (~6 papers) were also investigated for relevant data or information suitable for our manuscript. Topically relevant general review articles were excluded. Original clinical research, systematic reviews, and meta-analyses demonstrating statistical significance were dissected further with reference to criteria and outcomes involving continuous positive airway pressure (CPAP), bilevel positive airway pressure (BiPAP), chest injury activation protocol (ChIP), high flow nasal prongs (HFNP), hospital length of stay (HLOS), ICU length of stay (ILOS), intermittent positive pressure ventilation (IPPV), mechanical ventilator days (MVDs). This honing approach resulted in ~36 publications which informed the evidence behind our report.
Ethical approval was not required by our institution for this submission. However, in several (latter) parts of this paper, we have included observations and preliminary concepts/results from our use of SSRF in our trauma care practice at our hospital. Moreover, we have also included information about ongoing clinical studies linked to the information presented in this paper. Unfortunately, this paper does not have an adequate word count to fit into a submission under the “review” category. Therefore, this paper has been processed as a “brief report”.
This submission did not use the preferred reporting items for systematic reviews and meta-analysis (PRISMA) as it was not relevant. However, it is evident that we utilised a systematic approach for this submission. Moreover, we are currently preparing a systematic PRISMA-guided systematic review covering evidence-based management for elderly patients with rib fractures. This manuscript, which we expect to be submitted in a few weeks’ time, may have an adequate word count to fit into the “review category” as a systematic review.”
What are the strengths and weaknesses of your brief report?
- Towards this, we have added 5 new sections to our manuscript. Two of these sections are as follows.
- “Strengths of, and new information from our brief report
We analyse current thoracic trauma management algorithms (“bundles of care”) and evidence-based rib fracture therapy interventions (including SSRF), which focus on pain control, injury seriousness, and the anatomical extent of involvement. There is good evidence for the prognostic role of SSRF as part of a “bundle of care” in the setting of severe rib fractures such as ventilator-dependent patients and patients with flail chest. At our hospital, early SSRF (within 3 days) is recommended for patients with multiple rib fractures, flail chest, severe sternal fractures, extubation failure, and trauma patients requiring VATS.
We showed that blunt chest wall trauma and rib fractures are common in the elderly, with its associated mortality and morbidity far higher in the elderly when compared to that in younger patients [1]. Currently, the Gold Coast University Hospital is planning a multicentre RCT involving elderly patients with blunt chest wall trauma and rib fractures; and is recruiting investigators Australia-wide. Under current guidelines, pneumonia and contaminated fields are contraindications to SSRF. However, we have successfully conducted SSRF in several multi-trauma patients with contaminated wound fields and pneumonia.”
- “Weaknesses and research gaps identified in our brief report
Although good quality evidence exists for the utility of SSRF in patients with severe rib fractures and/or flail chest, we also reveal that SSRF use in flail chest treatment is minimal (0.7%) worldwide in these clinical settings [2]. Although several papers report that the use of SSRF in patients with multiple simple rib fractures lead to positive patient outcomes, the evidence from those papers comes mostly from retrospective and case-control trials, and require better quality evidence to substantiate this approach [3-6].
Although blunt chest wall trauma and rib fractures are common in the elderly, and their associated mortality and morbidity far higher, evidence for the clinical outcomes of SSRF in elderly rib fracture patients is sparse (anecdotal) or non-existent. This behoves the necessity of RCTs and prospective studies towards establishing evidence-based guidelines to manage this frail group of patients.
Although we presented the first ever retrospective study on outcomes of SSRF in multi-trauma patients with contaminated wound fields and positive sputum culture, our monocentric study had a small sample size.”
what is the take home massage from your brief report?
- Towards this, we have added 5 new sections to our manuscript.
- One of these new sections is entitled, “Take-home” messages from our brief report”. Another new section is entitled, “Conclusions”. Please see these 2 newly added segments below.
- “Take-home” points from our brief report
- The clinical impact of thoracic trauma depends on the severity of the trauma, the anatomical regions involved, and the age of patients.
- When initial interventions are unsatisfactory, VATs and SSRF must be considered considering the patient’s unique individual/holistic circumstances, clinical background, and prognostic expectations.
- Trauma centres and other healthcare providers should have updated evidence-based information about available management modalities - The best outcomes for individual patients are accomplished by combining various treatment approaches and individualising them according to the patient’s holistic situation and clinical condition.
- As part of rib fracture management pathways or “bundles of care”, early SSRF must be considered.
- Prospective studies and well-designed RCTs are required to confirm the benefits of SSRF in multiple simple rib fractures and in elderly patients with any sort of rib fracture.
- At our hospital, early SSRF is standard procedure for patients with multiple rib fractures, flail chest, severe sternal fractures, extubation failure, and trauma patients requiring VATS.
- In 2019, we presented the encouraging results of a small retrospective study on SSRF outcomes in multi-trauma patients with contaminated wound fields and positive sputum culture.”
- “Conclusions
The standard management of thoracic injury includes a multidisciplinary team approach with proper consideration of all potential avenues and treatment modalities (including SSRF) to give the best possible clinical outcome to the patient. Trauma centres and other healthcare providers should have updated evidence-based information about available management modalities. Multiple randomised control trials (RCTs), and meta-analyses show SSRF to be beneficial in flail chest patients resulting in reduced mortality and morbidity including shorter MVDs, HLOS, and ILOS; and lower incidence of infections, pneumonia, sepsis, and other complications. Prospective studies and well-designed RCTs are badly needed to confirm the benefits of SSRF in multiple simple rib fractures and in elderly patients with any sort of rib fracture. Such quality studies with good sample sizes are also essential to evaluate the outcome of SSRF in patients diagnosed with pneumonia or contaminated surgical fields.”
Updated references in our responses to reviewers’ comments
- Bulger, E.M.; Arneson, M.A.; Mock, C.N.; Jurkovich, G.J. Rib fractures in the elderly. The Journal of Trauma 2000, 48, 1040-1046; discussion 1046-1047, doi:10.1097/00005373-200006000-00007.
- Dehghan, N.; de Mestral, C.; McKee, M.D.; Schemitsch, E.H.; Nathens, A. Flail chest injuries: a review of outcomes and treatment practices from the National Trauma Data Bank. The Journal of Trauma and Acute Care Surgery 2014, 76, 462-468, doi:10.1097/ta.0000000000000086.
- Richardson, J.D.; Franklin, G.A.; Heffley, S.; Seligson, D. Operative fixation of chest wall fractures: an underused procedure? The American Surgery 2007, 73, 591-596; discussion 596-597.
- Nirula, R.; Allen, B.; Layman, R.; Falimirski, M.E.; Somberg, L.B. Rib fracture stabilization in patients sustaining blunt chest injury. Am Surg 2006, 72, 307-309, doi:10.1177/000313480607200405.
- Girsowicz, E.; Falcoz, P.E.; Santelmo, N.; Massard, G. Does surgical stabilization improve outcomes in patients with isolated multiple distracted and painful non-flail rib fractures? Interactive Cardiovascular and Thoracic Surgery 2012, 14, 312-315, doi:10.1093/icvts/ivr028.
- Campbell, N.; Conaglen, P.; Martin, K.; Antippa, P. Surgical stabilization of rib fractures using Inion OTPS wraps--techniques and quality of life follow-up. The Journal of Trauma 2009, 67, 596-601, doi:10.1097/TA.0b013e3181ad8cb7.

Reviewer 2 Report
The authors of the reviewed paper summarised thoracic cage anatomy and physiology, and thoracic trauma pathophysiology as well as the institutional clinical strategies and clinical pathway “bundles of care” available to reduce mortality and morbidity in patients with chest wall injury.
The topic of the paper is considered to be original enough and relevant to the field of critical care (the topic of the issue).
The authors conducted meticulous analysis of the current concepts in the the pathophysiology of chest wall injury, multimodal clinical pathways, and intervention strategies that include surgical stabilisation of rib fractures.
The accumulated data gives the clear prospective of the different approaches to the chest trauma including complications, mortality and others.
The paper is well structured and clearly presented.
The conclusions are supported by the published data, limitations and advantages of the paper are also presented.
The references are appropriate and mostly reflect the most recent publications.
Summarizing the findings the authors stress that further studies are required to evaluate the outcome of the SSRF inpatients diagnosed with pneumonia or contaminated surgical fields.
The paper makes good impression, contains important information and recommended for the publication in the present version.
Author Response
REVIEWER 2
The authors of the reviewed paper summarised thoracic cage anatomy and physiology, and thoracic trauma pathophysiology as well as the institutional clinical strategies and clinical pathway “bundles of care” available to reduce mortality and morbidity in patients with chest wall injury. The topic of the paper is considered to be original enough and relevant to the field of critical care (the topic of the issue). The authors conducted meticulous analysis of the current concepts in the the pathophysiology of chest wall injury, multimodal clinical pathways, and intervention strategies that include surgical stabilisation of rib fractures. The accumulated data gives the clear prospective of the different approaches to the chest trauma including complications, mortality and others. The paper is well structured and clearly presented. The conclusions are supported by the published data, limitations and advantages of the paper are also presented. The references are appropriate and mostly reflect the most recent publications. Summarizing the findings the authors stress that further studies are required to evaluate the outcome of the SSRF inpatients diagnosed with pneumonia or contaminated surgical fields. The paper makes good impression, contains important information and recommended for the publication in the present version.
- Thank you very much for your encouraging comments and positive assessment of our submitted manuscript.
Reviewer 3 Report
Dear authors,
"brief report" is described as follows
"Brief reports are short, observational studies that report preliminary results or a short complete study or protocol. Brief reports usually contain two figures and/or a table; however, the Materials and Methods sections should be detailed to ensure reproducibility of the presented work."
(check https://www.mdpi.com/about/article_types for further information)
The paper does not present any preliminary results and does not have any materials and methods section. In my opinion, it cannot be considered a "brief report". This kind of issue can not be solved with a simple revision, that's why I must recommend rejection.
Author Response
REVIEWER 3
Dear authors, "brief report" is described as follows - "Brief reports are short, observational studies that report preliminary results or a short complete study or protocol. Brief reports usually contain two figures and/or a table; however, the Materials and Methods sections should be detailed to ensure reproducibility of the presented work." (check https://www.mdpi.com/about/article_types for further information) The paper does not present any preliminary results and does not have any materials and methods section. In my opinion, it cannot be considered a "brief report". This kind of issue can not be solved with a simple revision, that's why I must recommend rejection.
- We have re-written the manuscript according to the instructions for “Brief Reports” in
- https://www.mdpi.com/journal/healthcare/instructions and
- https://www.mdpi.com/about/article_types
- We have added 5 new sections to our manuscript. One of these sections addresses this reviewer’s (Reviewer 3) concerns as follows -
“Materials and methods
Extensive searches were conducted using PubMed, Google Scholar, and government reports. Word searches with relevant combinations of words like chest trauma, chest injury, flail chest, randomised control trial, observation study, retrospective study, cross-sectional study, rib fracture; surgical stabilisation of rib fracture, thoracic injury, video-assisted thoracoscopic surgery, and outcomes. From more than 70 references, we identified ~32 papers which addressed management pathways for acute traumatic rib fractures involving surgical stabilisation. References from publications resulting from these sources (~6 papers) were also investigated for relevant data or information suitable for our manuscript. Topically relevant general review articles were excluded. Original clinical research, systematic reviews, and meta-analyses demonstrating statistical significance were dissected further with reference to criteria and outcomes involving continuous positive airway pressure (CPAP), bilevel positive airway pressure (BiPAP), chest injury activation protocol (ChIP), high flow nasal prongs (HFNP), hospital length of stay (HLOS), ICU length of stay (ILOS), intermittent positive pressure ventilation (IPPV), mechanical ventilator days (MVDs). This honing approach resulted in ~36 publications which informed the evidence behind our report.
Ethical approval was not required by our institution for this submission. However, in several (latter) parts of this paper, we have included observations and preliminary concepts/results from our use of SSRF in our trauma care practice at our hospital. Moreover, we have also included information about ongoing clinical studies linked to the information presented in this paper. Unfortunately, this paper does not have an adequate word count to fit into a submission under the “review” category. Therefore, this paper has been processed as a “brief report”.
This submission did not use the preferred reporting items for systematic reviews and meta-analysis (PRISMA) as it was not relevant. However, it is evident that we utilised a systematic approach for this submission. Moreover, we are currently preparing a systematic PRISMA-guided systematic review covering evidence-based management for elderly patients with rib fractures. This manuscript, which we expect to be submitted in a few weeks’ time, may have an adequate word count to fit into the “review category” as a systematic review.”
Reviewer 4 Report
Although I have no idea on "Brief Report", this would be a kind of essay rather than scientific report. Moreover, it is redundant with lots of duplicates.
Brief reports are short, observational studies that report preliminary results or a short complete study or protocol. Brief reports usually contain two figures and/or a table; however, the Materials and Methods sections should be detailed to ensure reproducibility of the presented work." This paper does not present any preliminary results and does not have any materials and methods section.
Author Response
REVIEWER 4
Although I have no idea on "Brief Report", this would be a kind of essay rather than scientific report. Brief reports are short, observational studies that report preliminary results or a short complete study or protocol. Brief reports usually contain two figures and/or a table; however, the Materials and Methods sections should be detailed to ensure reproducibility of the presented work."
This paper … does not have any materials and methods section.
- We have added 5 new sections to our manuscript. One of these sections addresses this reviewer’s (Reviewer 3) concerns as follows -
“Materials and methods
Extensive searches were conducted using PubMed, Google Scholar, and government reports. Word searches with relevant combinations of words like chest trauma, chest injury, flail chest, randomised control trial, observation study, retrospective study, cross-sectional study, rib fracture; surgical stabilisation of rib fracture, thoracic injury, video-assisted thoracoscopic surgery, and outcomes. From more than 70 references, we identified ~32 papers which addressed management pathways for acute traumatic rib fractures involving surgical stabilisation. References from publications resulting from these sources (~6 papers) were also investigated for relevant data or information suitable for our manuscript. Topically relevant general review articles were excluded. Original clinical research, systematic reviews, and meta-analyses demonstrating statistical significance were dissected further with reference to criteria and outcomes involving continuous positive airway pressure (CPAP), bilevel positive airway pressure (BiPAP), chest injury activation protocol (ChIP), high flow nasal prongs (HFNP), hospital length of stay (HLOS), ICU length of stay (ILOS), intermittent positive pressure ventilation (IPPV), mechanical ventilator days (MVDs). This honing approach resulted in ~36 publications which informed the evidence behind our report.
Ethical approval was not required by our institution for this submission ... Unfortunately, this paper does not have an adequate word count to fit into a submission under the “review” category. Therefore, this paper has been processed as a “brief report”.
This submission did not use the preferred reporting items for systematic reviews and meta-analysis (PRISMA) as it was not relevant. However, it is evident that we utilised a systematic approach for this submission. Moreover, we are currently preparing a systematic PRISMA-guided systematic review covering evidence-based management for elderly patients with rib fractures. This manuscript, which we expect to be submitted in a few weeks’ time, may have an adequate word count to fit into the “review category” as a systematic review.”
===================================
This paper does not present any preliminary results and does not have any materials and methods section.
- In several (latter) parts of this paper, we have included observations and preliminary concepts/results from our use of SSRF in our trauma care practice at our hospital. Moreover, we have also included information about ongoing clinical studies linked to the information presented in this paper.
===================================
Moreover, it is redundant with lots of duplicates.
- Please specify what segments of our original manuscript are “…redundant with lots of duplicates”, and we will certainly amend those segments.
===================================
